# Prospective Evaluation of a New Liquid-Type Rapid Urease Test Kit for Diagnosis of *Helicobacter pylori*

**DOI:** 10.3390/diagnostics14070700

**Published:** 2024-03-27

**Authors:** Seung Han Kim, Kyeong Ah Kim, Moon Kyung Joo, Hannah Lee, Jun-Won Chung, Sung-Cheol Yun, Seon Tae Kim

**Affiliations:** 1Department of Gastroenterology, Korea University Guro Hospital, Seoul 08308, Republic of Korea; kimseunghan09@gmail.com (S.H.K.); latyrx@korea.ac.kr (M.K.J.); 2Department of Internal Medicine, Gachon University Gil Medical Center, Incheon 21565, Republic of Korea; ruddk97@hanmail.net (K.A.K.); everagape@gmail.com (H.L.); 3Department of Clinical Epidemiology and Biostatistics, University of Ulsan College of Medicine, Asan Medical Center, Seoul 05505, Republic of Korea; ysch97@amc.seoul.kr; 4Department of Otolaryngology-Head & Neck Surgery, Gachon University Gil Medical Center, Incheon 21565, Republic of Korea; kst2383@gilhospital.com

**Keywords:** *Helicobacter pylori*, rapid urease test, liquid-type medium

## Abstract

Background/Aims: Rapid and accurate diagnostic tools are essential for the timely recognition of Helicobacter pylori (*H. pylori*) in clinical practice. The rapid urease test (RUT) is a comparatively accurate and time-saving method recommended as a first-line diagnostic test. The primary objective of conducting the RUT is to obtain rapid results, thus enabling the initiation of an eradication therapy based on clarithromycin resistance testing. This study aimed to assess the reaction time and accuracy of a new liquid-type RUT. Method: In this prospective study, consecutive dyspeptic or check-up patients referred to our clinic for endoscopy were assessed to evaluate the rapidity and accuracy of a novel liquid-type RUT (Helicotest^®^, WON Medical, Bucheon, Republic of Korea) compared with another commercial RUT kit (HP kit, Chong Kun Dang, Seoul, Republic of Korea) and a real-time quantitative PCR-based assay (Seeplex^®^ H.pylori-ClaR Detection, Seegene, Republic of Korea). RUTs were analyzed at 10 min, 30 min, 60 min, and 120 min. Results: Among the 177 enrolled patients, 38.6% were infected with *H. pylori*. The positivity rates of the liquid-type RUT were 26.1, 35.8, 39.2%, and 41.5% at 10, 30, 60, and 120 min, respectively. When compared with the HP kit test, the time needed to confirm positivity was significantly reduced by 28.6 min (95% CI, 16.60–39.73, *p* < 0.0001). Helicotest^®^ had a greater accuracy (96.02 ± 1.47), sensitivity (98.53 ± 1.46) and NPV (99.03 ± 0.97) compared to the HP kit. Conclusions: Compared to the commonly used RUT, the new liquid-type RUT presented faster and reliable results. Such findings could improve *H. pylori* treatment outcomes, particularly in outpatient clinical settings.

## 1. Introduction

*Helicobacter pylori* (*H. pylori*) is a spiral and gram-negative bacterium that infects the human stomach. *H. pylori* is related to various gastric diseases and affects the development of gastric ulcers and adenocarcinoma [1,2,3,4,5,6].

*H. pylori* treatment involves a combination of robust acid inhibitors with various combinations of antibiotics and/or bismuth. However, the frequency of antibiotic resistance in *H. pylori* has gradually increased over the past decades [7,8,9,10,11,12,13].

Antibiotic resistance of *H. pylori* strains is the main cause of failure in eradication treatment and has varying degrees of prevalence according to the country and over time. The *H. pylori* eradication rate has continued to decrease due to the increasing antibiotic resistance of *H. pylori*.

To overcome antibiotic resistance, various antibiotic combinations have been studied, and according to the recent guidelines, if resistance is noted based on antibiotic resistance testing, bismuth-based quadruple therapy or PPI-amoxicillin-metronidazole (PAM) treatment can be considered. Consequently, conducting sensitivity testing for the major antibiotics used for *H. pylori* treatment and setting appropriate treatment strategies has become increasingly important.

In order to quickly confirm *H. pylori* infection and perform antibiotic resistance testing accordingly, it has become important to have an appropriate, rapid, outpatient-based test method that quickly leads to resistance testing. However, conventional testing methods, such as culture, biopsy, and urea breath tests, take too much time from confirmation of *H. pylori* to susceptibility testing.

Among the various diagnostic tests for the detection of *H. pylori* infection, the rapid urease test (RUT) is a comparatively precise and rapid method and is suggested as a first-line diagnostic method [14,15,16,17,18,19]. As a “test and treat” approach was recommended for *H. pylori*-infected patients in the Maastricht V consensus [7], the rapidity and accuracy of the diagnostic test for *H. pylori* are valuable. Also, the recently published Maastricht VI consensus suggested the need for an antibiotic resistance test prior to *H. pylori* eradication treatment, including clarithromycin [20]. If positive results are confirmed by performing RUT and the tissue is immediately used for antibiotic resistance testing, as suggested in the Maastricht VI consensus, the burden of having to perform the test again for resistance testing can be reduced, and patient compliance can be improved by shortening the time.

RUT is a diagnostic method that uses stomach biopsy tissue, is relatively simple, and has good sensitivity and specificity, making it highly clinically useful. Among these, the CLO kit (Halyard, Alpharetta, GA, USA) and HP kit (Chong Kun Dang, Seoul, Republic of Korea) are mainly used in Korea. Among these, the HP kit is mentioned in few studies, and the positivity rate of the device varies depending on the various clinical situations. Additionally, diagnostic reagents have the disadvantage of requiring at least 20 min to 2 h to read the final results.

The main challenge in performing the RUT is to obtain a rapid and reliable result. To address this challenge, we developed a new liquid-type RUT kit (Helicotest^®^, WON Medical, Republic of Korea) for diagnosing *H. pylori* infection. In a previous study [21], we conducted an in vitro investigation using two *H. pylori* strains (*H. pylori* ATCC 43504 and 700392), and the novel liquid-type RUT kit (Helicotest^®^) presented the rapidest response speed compared to the other three types of RUT kits available on the market.

Helicotest^®^ has the advantage of being able to read the final results within 5 to a maximum of 30 min. It is expected to have an excellent sensitivity and specificity performance compared to control diagnostic reagents and also to reduce reading time.

This clinical study aimed to assess the response time and accuracy of the novel liquid-type RUT.

## 2. Materials and Methods

### 2.1. Study Population

Between January 2022 and December 2022, 177 consecutive patients were enrolled in two tertiary hospital. Patients who underwent upper gastrointestinal endoscopy for dyspepsia or for screening purposes were prospectively included. Patients were excluded if they had undertaken *H. pylori* eradication therapy in the past, had been prescribed anbiotics or proton pump inhibitors within the previous 4 weeks, had a history of previous gastric surgery, were pregnant or lactating, were under the age of 18 years, or if they were deemed ineligible for participation in the trial by the researchers.

All patients signed an informed consent form before the study was initiated. Ethical approval was obtained from the institutional review board of Korea University Guro Hospital (IRB No. 2021GR0058) and from the institutional review board of Gachon University Gil Medical Center (IRB No. 2021GR0058). After obtaining consent for the study from 177 patients who met the inclusion criteria, tissue samples were collected during endoscopy. However, as a result, samples from 176 people were collected, excluding one patient who withdrew consent during the study.

### 2.2. Tissue Collection

During upper gastrointestinal endoscopy, three tissue samples were gathered from each patient. We obtained samples from the lesser side of the gastric antrum and the great curvature of the gastric corpus, preferably from non-atrophic areas, using biopsy forceps (BioCeps^®^, Diagmed Healthcare, North Yorkshire, UK) with a catheter diameter of 2.4 mm and length 160 cm. A total of 6 tissues were collected, and 2 tissues each from the gastric antrum and body were subjected to the Helicotest^®^ (WON Medical, Bucheon, Republic of Korea), 2 tissues to the HP kit (Chong Kun Dang, Seoul, Republic of Korea), and 2 tissues to the real-time quantitative PCR-based assay (Seeplex^®^ H.pylori-ClaR Detection, Seegene, Seoul, Republic of Korea). If the gastric tissue was collected for histological evaluation, the biopsy tissue was not placed directly into the formalin sample container but was transferred to a small piece of paper and placed into the sample container.

### 2.3. Diagnostic Tests for H. pylori Infection

We conducted two types of RUTs, using two different kits, in addition to PCR tests. The Helicotest^®^ (WON Medical, Republic of Korea), the new liquid-type RUT test kit, was made by combining sodium dihydrogen phosphate monohydrate (NaH_2_PO_4_/H_2_O), phenol red, sodium acetate (C_2_H_3_O_2_Na), and urea. When urea is hydrolyzed by urease secreted by *H. pylori*, the color of the medium changes to orange, red or purple. The change in color is determined using a colorimetric method.

The Helicotest^®^ RUT kit was maintained at room temperature for at least 30 min before use and was observed at room temperature, even after the stomach tissue sample was stored. For all RUT kits, color changes were observed at 10 min, 30 min, 1 h, and 2 h. Helicotest^®^ can be read by visual observation after 5 min; a red, orange, or pink color indicates a positive result. Conversely, a yellow or unchanged color indicates a negative result.

The HP kit (Chong Kun Dang, Seoul, Republic of Korea) was used as the control RUT kit. Results were observed with the naked eye after 10 min. A positive result was considered positive if there was a color change from orange to pink.

A real-time quantitative PCR-based assay (Seeplex^®^ H.pylori-ClaR Detection, Seegene, Seoul, Republic of Korea) was performed. Antibiotic susceptibilities to clarithrymycin were tested by real-time quantitative PCR-based assay.

### 2.4. Confirmation of H. pylori Infection

*H. pylori* infection was diagnosed when two or more of the three tests (the PCR test and two types of RUT (Helicotest^®^ and HP kit)) were positive. A patient was considered non-infected if all tests were negative or only one was positive.

### 2.5. Comparison with Other Commercial Kit

Helicotest^®^ (WON Medical, Bucheon, Republic of Korea) was compared with the HP kit (Chong Kun Dang, Seoul, Republic of Korea), another semisolid (agar gel) commercial kit, and the results were expressed as the recorded color changing time to compare reaction rates over time.

### 2.6. Statistical Analysis

To calculate the number of research subjects, we assumed that the 2 h positivity rate of the HP kit (Chong Kun Dang, Seoul, Republic of Korea) was expected to be 70% and the positivity rate of the newly developed Helicotest^®^ (WON Medical, Bucheon, Republic of Korea) was expected to be 85%, the testers’ *H. pylori* positivity rate was 70%, the marginal error was 0.05, the power was at 80%, and 160 patients were needed. If the proportion discordant rate of the HP kit (Chong Kun Dang, Seoul, Republic of Korea) is set to 30% and the dropout rate is set to 10%, samples from 176 patients are needed.

The chi-square test and Fisher’s exact test were used for statistical analysis. Descriptive statistics were used to present measurable data, which are expressed as mean ± SD. The sensitivity, specificity, positive predictive value (PPV), and negative predictive value (NPV) were calculated when two out of the three tests yielded positive results.

When both the test and control RUT kits tested positive, the first discoloration time was presented as a descriptive statistic. This included the number of subjects, mean, standard deviation, median value, minimum value, and maximum value for each method. The statistical significance of the differences between both kits was tested at a significance level of 0.05, using an independent two-sample *t*-test or Wilcoxon rank-sum test.

## 3. Results

Of the 176 prospectively enrolled patients, the average age was 58.6 ± 12.9 years old, and males accounted for 46%. Endoscopic findings included gastric ulcers in 14 patients, duodenal ulcers in 7, gastric adenoma in 15, and early gastric cancer in 9 patients. Other variables are listed in Table 1.

Among these patients, 38.6% were infected with *H. pylori*. The results of agreement between the test device and the HP kit were confirmed as an overall agreement of 88.07%, positive agreement of 98.15%, and negative agreement of 83.61%. Among these, the reason why the negative agreement is relatively lower than the positive agreement is because the HP kit showed negative results for all 23 positive confirmed samples, which is 29.87% of the samples confirmed positive through confirmation test. Helicotest^®^ showed negative results in 12 out of 23 positive confirmed samples. Due to the different results of the 11 samples, the negative agreement was analyzed to be relatively lower than the positive agreement, and also, due to the low negative agreement, the overall agreement was analyzed to be low.

The positivity rates of the new RUT were 26.1, 35.8, 39.2%, and 41.5% at 10, 30, 60, and 120 min, respectively (Table 2). Results from comparing the number of samples determined to be positive over time showed that Helicotest^®^ showed a positive color change in 46 confirmed positive samples (59.74%) in 10 min, while the HP kit showed a positive color change in only 10 positive samples (12.98%).

The Helicotest^®^ showed a positive color in 63 samples (81.82%) of the confirmed positive samples, at 30 min, while the HP kit showed a color change in 28 positive samples (36.36%). After 2 h of reaction time, the Helicotest^®^ identified 73 samples (94.80%) of the positive samples as positive, while the HP kit identified only 70.13% of the samples as positive.

When compared with the HP kit, Helicotest^®^ showed a faster positive reaction rate, and the difference according to the time to confirm positivity was 28.6 min (95% CI, 16.60 to 39.73, *p* < 0.0001). The accuracy, sensitivity, specificity, PPV, and NPV of the three *H. pylori* tests are shown in Table 3. Helicotest^®^ had a greater accuracy (96.02 ± 1.47), sensitivity (98.53 ± 1.46) and NPV (99.03 ± 0.97) compared to the HP kit.

## 4. Discussion

This study aims to evaluate the sensitivity, specificity, and correlation of a test device for clinical performance testing using human stomach tissue samples at two tertiary medical institutions.

In the current study, Helicotest^®^, a newly developed liquid-type RUT kit, showed a faster positive reaction time when compared with the HP kit, a commonly used RUT kit in South Korea. Helicotest^®^ is an in vitro diagnostic device that qualitatively tests the urease activity of *H. pylori* and confirms the presence of *H. pylori* in human stomach tissue. Clinical specificity and sensitivity were evaluated, and concordance was assessed with the approved reference medical device, the HP kit. After obtaining consent for the study from patients who met the inclusion criteria, gastric tissue samples were collected during endoscopy, resulting in 176 samples being collected. A clinical efficacy evaluation was performed on 176 gastric biopsy tissue samples.

In the primary efficacy evaluation, the 176 test results were compared with the test results of the HP kit to confirm a correlation. As a result, the positive match rate with the HP kit was 98.15%, negative agreement rate was 83.61%, Cohen’s kappa value was 0.7445, and it was confirmed that it met the validity evaluation criteria for clinical performance of 0.6 or higher.

Most *H. pylori*-positive cases are asymptomatic, but continued colonization can cause a variety of gastric and extragastric diseases [22]. Stomach disorders caused by infection begin with gastritis, and some patients with chronic atrophic gastritis and intestinal metaplasia develop gastric dysplasia, which is a precursor of gastric cancer [23,24,25,26]. Therefore, *H. pylori* is considered important in gastric carcinogenesis and was classified as a human cancer-causing agent by the World Health Organization’s International Agency for Research on Cancer in 1994 [27]. *H. pylori* is a first-class carcinogen, and potential cancers include gastric mucosa-associated lymphoid tissue (MALT) lymphoma and gastric cancer. *H. pylori* infection accounts for approximately 89% of all gastric cancers and is associated with the occurrence of 5.5% of all cancer cases worldwide [28]. Therefore, in patients infected with *H. pylori*, proper assessment and efficient management are important to prevent gastric cancer and its related complications.

Antibiotic resistance of *H. pylori* strains is the main cause of failure in eradication treatment and has varying degrees of prevalence according to the country and over time. The *H. pylori* eradication rate has continued to decrease due to the increasing antibiotic resistance of *H. pylori*. With the increasing antibiotic resistance rate of *H. pylori*, the Maastricht VI consensus recommended an antibiotic resistance test before *H. pylori* eradication treatment including clarithromycin [20].

From the results of two recently published case-control studies, the per-protocol analysis presented that the eradication rates for 7 days of customized treatment using bismuth quadruple therapy, PAM, or standard triple therapy were 91.8% and 94.3%, respectively. This was higher than the values of 72.1% and 76.5% of the empirical standard therapy group [12,29]. In particular, the cost of customized treatment was almost equal to the cost of standard empirical therapy for 14 days, so it was not inferior in terms of cost-effectiveness [30].

Nevertheless, performing the resistance test at a later stage, following the initial assessment of the presence of *H. pylori*, might result in delays that could reduce patient compliance, thus compromising the eradication treatment of *H. pylori*. Therefore, Maastricht VI also proposed an antibiotic resistance test through biopsy tissue reuse, presenting potential advantages in the context of the RUT test. This approach is very useful in terms of cost reduction and lessening the burden on patients and physicians. It allows a rapid check of *H. pylori* positivity after RUT and the subsequent reuse of the same tissue for antibiotic resistance tests in the case of a positive result.

RUT is a widely used diagnostic test based on the characteristic of *H. pylori* that produces urease. The produced urease converts urea into ammonia and increases the pH, consequently changing the color of the medium in the RUT kit. Although the RUT is a simple method, it takes a considerable amount of time to provide accurate results, and the response times vary from 5 min to 24 h. Therefore, various media have been developed to obtain rapid results, including gels, paper, and liquids. The properties of the media and density of *H. pylori* affect the response speed and accuracy of the diagnostic device. Therefore, the development of a medium that reacts accurately and quickly with low-density bacteria will provide greater clinical usefulness.

We developed a novel liquid-type RUT kit for the diagnosis of *H. pylori* infection. Helicotest^®^ presented a faster positive reaction rate, with a notable reduction of 28.6 min in the time required to confirm positivity, when compared with the control RUT kit. The positivity rates of the new RUT were 26.1, 35.8, 39.2%, and 41.5% at 10, 30, 60, and 120 min, respectively.

The rapid response of this new RUT kit will increase patient compliance to *H. pylori* treatment based on the reduction in time and the high adaptation rate to the resistance test compared to other diagnostic methods. This will increase the ability to overcome the escalating issue of antibiotic resistance in *H. pylori*. Considering the rapidity and high sensitivity of the liquid-type RUT, particularly Helicotest^®^, it also could be advantageous in terms of cost-effectiveness.

However, it is essential to acknowledge the limitations of this study. Specifically, as Helicotest^®^ is a test kit in liquid form, storage and transportation methods can affect the reaction time and impact the results. However, in this study, efforts were made to maintain the same storage and temperature conditions, specifically maintaining the kit at room temperature for a certain period before conducting the test. Another limitation was the comparison of the Helicotest^®^ with only one RUT test kit. Moreover, as the sample size was relatively small, the usefulness of the Helicotest^®^ should be addressed in a large-scale clinical study. Also, because the study was conducted on patients undergoing upper gastrointestinal endoscopy for screening purposes, patients with various past histories (gastric ulcer, early gastric cancer, MALT lymphoma) were included. Although there is a possibility that the use of probiotics may affect the composition of gastric microbiome, this was not investigated and may also be a limitation of the study.

## 5. Conclusions

In conclusion, compared to the commonly used RUT, the new liquid-type RUT presented a rapid result. This could improve *H. pylori* treatment outcomes in outpatient clinical settings. Future studies are needed to demonstrate its usefulness in actual clinical practice.

## Figures and Tables

**Table 1 diagnostics-14-00700-t001:** Demographic and clinical characteristics (N = 176).

Variable		N	%
Age (Mean ± SD)		58.6 ± 12.9	
Gender (Male)		85	48.2
Current smoking		40	22.7
Current alcohol consumption		70	39.8
Previous GI disease	0. None	81	46.0
	1. GU	14	8.0
	2. DU	7	4.0
	3. GU+DU	1	0.6
	4. Adenoma	15	8.5
	5. EGC	9	5.1
	6. Gastric MALT lymphoma	1	0.6
	7. Others	58	33.0
Comorbidity	0. None	73	
	1. Hypertension	55	31.3
	2. Diabetes	28	15.9
	3. Ischemic heart disease	7	4.0
	4. Liver cirrhosis	2	1.1
	5. Malignancy	13	7.4
	6. Others	35	19.9
PCR Hp	Positive	77	43.8
Mutation	None	148	84.1
	2. A2142G positive	1	0.6
	3. A2143G positive	27	15.3

N: number of patients; SD: standard deviation; GI: gastrointestinal; GU: gastric ulcer; DU: duodenal ulcer; EGC: early gastric cancer; MALT: mucosa-associated lymphoid tissue. Values are presented as mean (standard deviation) for age.

**Table 2 diagnostics-14-00700-t002:** Response speed of two RUT tests.

Variable		N	%
Helicotest^®^	10 min	46	26.1
	30 min	63	35.8
	1 h	69	39.2
	2 h	73	41.5
	Mean ± SD (min)	24.8 ± 27.44	
HP kit	10 min	10	5.7
	30 min	28	15.9
	1 h	43	24.4
	2 h	54	30.7
	Mean ± SD (min)	53.0 ± 38.4	
Confirmation of *H.pylori* infection		68	38.6

N: number of patients; SD: standard deviation.

**Table 3 diagnostics-14-00700-t003:** Validity of three *H. pylori* tests.

	Accuracy	Sensitivity	Specificity	PPV	NPV	Kappa
HP kit	92.05 ± 2.04	79.41 ± 4.90	100 ± 0.65	100 ± 1.29	88.52 ± 2.89	0.8256 ± 0.044
Helicotest^®^	96.02 ± 1.47	98.53 ± 1.46	94.44 ± 2.20	91.78 ± 3.21	99.03 ± 0.97	0.9172 ± 0.030
PCR	92.61 ± 1.97	97.06 ± 2.05	89.81 ± 2.91	85.71 ± 3.99	97.98 ± 1.41	0.848 ± 0.040

PPV: positive predictive value; NPV: negative predictive value. Values are presented as mean (standard deviation).

## Data Availability

The data presented in this study are available on request from the corresponding author due to privacy reasons.

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
