# Peer review of "Prospective Evaluation of a New Liquid-Type Rapid Urease Test Kit for Diagnosis of Helicobacter pylori"

_diagnostics, 2024, doi:10.3390/diagnostics14070700_

Round 1
Reviewer 1 Report
Comments and Suggestions for Authors
The manuscript presents evaluationn of a new RUT for the rapid diagnosis of H pylori infection Although the methodology, statistics and presentation of the material is sound, I have several theoretical and prascticval concerns as follows:
1. Trivial data on H pylori discovery and prevalence, etc. are not necesssry to understand the manuscript (see Introduction and Discussion section) ,
2. The exact chemical composition of the new test iincluding dosage of reagents) is incompletely presented (if not patented)
3. Were there bioptic samples collected for histological exam too and if yes, was the forceps for RUT used firstly? (formalin kills the bacteria!)
4. Were the RUT examinaed at room temperature of body temperature?
5. Hoiw many particles were put in nthe RUT recipient, 1 or 2? (increasing the no of particles increassdes the accuracy)
6. Are the 28 minutes time gain with the new test really important for daily prsactice?
7.The results from antrum and coerporeal biopsies should be presented separately (accuracy, sensitivity, specificity NPV,PPV))
8. Including dyspepsia, pepticnulcer, EGA and MALT lymohoma is the same group is inappropriate because the prevalence of H pylori in alll these conditions are different
9. Do you use the samer bioptic mysterial for histology and antimicrobial resistence determination?
10. Probiotics can either redduce urease activie and thered are sztrainf of gastric microbime having urease activity, generating either false negativedd or false positive results.
Please comment for problemsd mentioned above.
Author Response
Reviewer 1
The manuscript presents evaluationn of a new RUT for the rapid diagnosis of H pylori infection Although the methodology, statistics and presentation of the material is sound, I have several theoretical and prascticval concerns as follows:
Comment 1: Trivial data on H pylori discovery and prevalence, etc. are not necesssry to understand the manuscript (see Introduction and Discussion section),
Response 1: We thank the reviewer for the valuable comment. Introduction and discussion section were revised as like your suggestion.
Comment 2: The exact chemical composition of the new test iincluding dosage of reagents) is incompletely presented (if not patented)
Response 2: We know the importance of your suggestion, however, since it has been patented, it cannot be disclosed in this manuscript.
Comment 3: Were there bioptic samples collected for histological exam too and if yes, was the forceps for RUT used firstly? (formalin kills the bacteria!)
Response 3: We thank the reviewer for the worthy comment. We also know concerns about that. Therefore, in this study, the biopsy tissue is not placed directly into the formalin sample container, but is transferred to a small piece of paper and placed into the sample container. In response to this comment, we have added the following text to the MATERIALS and METHODS section (page 3, lines 108-110): “If the gastric tissue was collected for histological evaluation, the biopsy tissue was not placed directly into the formalin sample container, but is transferred to a small piece of paper and placed into the sample container.”
Comment 4. Were the RUT examinaed at room temperature of body temperature?
Response 4: The Helicotest® RUT kit was maintained at room temperature for at least 30 min before use and was observed at room temperature. It is descripted in the MATERIALS and METHODS section (page 3, lines 118-120): “The Helicotest® RUT kit was maintained at room temperature for at least 30 min before use and was observed at room temperature, even after the stomach tissue sample was stored.”
Comment 5. Hoiw many particles were put in nthe RUT recipient, 1 or 2? (increasing the no of particles increassdes the accuracy)
Response 5: A total of 6 gastric tissues per one patient were collected for this study, and 2 tissues each from the gastric antrum and body were subjected to the Helicotest® (WON Medical, Republic of Korea). It is descripted in the MATERIALS and METHODS section (page 3, lines 136-137): 2 tissues each from the gastric antrum and body were subjected to the Helicotest® (WON Medical, Republic of Korea)”
Comment 6. Are the 28 minutes time gain with the new test really important for daily prsactice?
Response 6: The rapid response of this new RUT kit could increase patient compliance to H. pylori treatment based on the reduction in time and the high adaptation rate to the resistance test compared to other diagnostic methods. This could increase the ability to overcome the escalating issue of antibiotic resistance in H. pylori. Considering the rapidity and high sensitivity of the liquid-type RUT, particularly Helicotest®, it also could be advantageous in terms of cost-effectiveness. It is descripted in the DISCUSSION section (page 7, lines 262-267): “The rapid response of this new RUT kit will increase patient compliance to H. pylori treatment based on the reduction in time and the high adaptation rate to the resistance test compared to other diagnostic methods. This will increase the ability to overcome the esca-lating issue of antibiotic resistance in H. pylori. Considering the rapidity and high sensi-tivity of the liquid-type RUT, particularly Helicotest®, it also could be advantageous in terms of cost-effectiveness.”
Comment 7.The results from antrum and coerporeal biopsies should be presented separately (accuracy, sensitivity, specificity NPV,PPV))
Response 7: We thank the reviewer for the valuable comment. But, samples such as PCR were not performed by separating the antrum and body, and in the case of RUT, if only one of the two site was positive, it was considered as positive, so it was not calculated separately.
Comment 8. Including dyspepsia, pepticnulcer, EGA and MALT lymohoma is the same group is inappropriate because the prevalence of H pylori in alll these conditions are different
Response 8: We thank the reviewer for the worthy comment. Your comments on patient population are reasonable. However, it was about past history, not current disease, and patients who had previously undergone HP eradication were excluded. Additionally, similar proportions of the past history were assigned to each group. The population related descriptions were added as limitation of this study in DISCUSSION section (page 7, lines 275-277): “Also, because the study was conducted on patients undergoing upper gastrointestinal endoscopy for screening purposes, patients with various past histories (gastric ulcer, early gastric cancer, MALT lymphoma) were included.”
Comment 9. Do you use the samer bioptic mysterial for histology and antimicrobial resistence determination?
Response 9: We did not use the same tissue for histologic evaluation and antibiotic resistance assessment. Antibiotic susceptibilities for clarithromycin was tested by real-time quantitative PCR-based assay and it was added in MATERIAL and METHODS section (page 4, lines 130-131): “Antibiotic susceptibilities for clarithromycin was tested by real-time quantitative PCR-based assay.”
Comment 10. Probiotics can either redduce urease activie and thered are sztrainf of gastric microbime having urease activity, generating either false negativedd or false positive results.
Response 10: We thank the reviewer for the valuable comment. Your comments on probiotics use is correct. The probiotics related descriptions were added as limitation of this study in DISCUSSION section (page 7, lines 279-281): “Although there is a possibility that the use of probiotics may affect the composition of gastric microbiome, but this was not investigated and this may also be a limitation of the study.”

Reviewer 2 Report
Comments and Suggestions for Authors
See the attached file

Author Response
Reviewer 2
This study assesses the reaction time and accuracy of a new liquid-type RUT. The author evaluates an important diagnostic tool that has received little attention in national studies. The introduction clearly states the topic and the study's investigation strategy is clear, which influenced the article's results. However, the discussion parts did not clearly discuss the study results. Additionally, I have some comments that need to be edited to improve the manuscript.
Comment 1: In the methodology section, line 88, the authors wrote that the total number of patients enrolled in the study was 177. However, in the results section, line 159, they wrote 176!
Response 1: After obtaining consent for the study from 177 patients who met the inclusion criteria, tissue samples were collected during endoscopy. However, as a result, samples from 176 people were analyzed for the study excluding one patient who withdrew consent during the study. It is descripted in the MATERIALS and METHODS section (page 2, lines 95-98): “After obtaining consent for the study from 177 patients who met the inclusion criteria, tis-sue samples were collected during endoscopy. However, as a result, samples from 176 people were collected excluding one patient who withdrew consent during the study.”
Comment 2: Methodology section, lines 134-135, What does it mean’’A patient was considered 134 non-infected if all tests were negative or only one was positive’’.
Response 2: We performed two types of RUT (Helicotest® and HP kit) and the PCR test for the confirmation of H. pylori infection. H. pylori infection was diagnosed when two or more of the three test (the PCR test and two types of RUT) are positive. Therefore, if one test was positive or all were negative, it was defined as H. pylori negative.
Comment 3: In the results section: Table 1, the total number of patients classified according to the Previous GI disease is more than 176?!
Response 3: This is because there are patients with two or more previous GI diseases.
Comment 4: Similarly, Table 1 shows the total number of comorbidities and Mutations that are not correlated with the sample size.
Response 4: The total number of mutations is 176, which is consistent with the number of study subjects, and the total number of comorbidities is also because there are patients with two or more comorbidities.
Comment 5: The authors have not provided information about the outcome of children who were admitted to the ICU or had a fever but were not prescribed antibiotics. It is crucial to assess whether administering antibiotics to those at high risk would reduce the likelihood of co-infection or secondary bacterial infection, which could exacerbate the condition of hospitalized children.
Response 5: This question is judged to be unrelated to the research content of our paper and requires confirmation from the editor.
Comment 6:- Table 2: what does the Golden standard refer to?
Response 6: We thank the reviewer for the good comment. We changed "Golden standard" to "confirmation of H.pylori infection" because of the potential for confusion. The contents of Table 2 have been modified.
Comment 7: In the discussion, lines 215-245, need to be deleted as the authors provide background knowledge on H. pylori, which would be better suited for the introduction section rather than the discussion, as the discussion should focus on the comparison of your data.
Response 7: We thank the reviewer for the comment. In response to this comment, we modified the text.
Comment 8:- There is a lack of comprehensive scientific discussion of the data as authors expected to discuss their data in general as well as some important findings, such as the reason for having negative results with Helicotest® although PCR was positive. Also, the PCR accuracy was 92.61 ± 1.97, but Helicotest® shows 96.02 ± 1.47
Response 8: We thank the reviewer for the worthy comment. We totally agree with the reviewers. However, currently, there is no 100% precise diagnostic test for H. pylori detection, and test results are sometimes mixed. The PCR method is very accurate, but there are cases of false positives. It is thought that accuracy might vary depending on the test.
